# Genome-Wide Evolutionary Characterization and Expression Analysis of Major Latex Protein (MLP) Family Genes in Tomato

**DOI:** 10.3390/ijms241915005

**Published:** 2023-10-09

**Authors:** Zhengliang Sun, Liangzhe Meng, Yuhe Yao, Yanhong Zhang, Baohui Cheng, Yan Liang

**Affiliations:** College of Horticulture, Northwest A&F University, Xianyang 712100, China; sunzhengliang99@163.com (Z.S.); mengliangzhe@nwafu.edu.cn (L.M.); 17860369697@163.com (Y.Y.); 2021055322@nwafu.edu.cn (Y.Z.); 2021055286cbh@nwafu.edu.cn (B.C.)

**Keywords:** *Solanum lycopersicum*, MLP, genome-wide identification, evolutionary relationships, plant growth, stress response

## Abstract

Major latex proteins (MLPs) play a key role in plant response to abiotic and biotic stresses. However, little is known about this gene family in tomatoes (*Solanum lycopersicum*). In this paper, we perform a genome-wide evolutionary characterization and gene expression analysis of the MLP family in tomatoes. We found a total of 34 SlMLP members in the tomato genome, which are heterogeneously distributed on eight chromosomes. The phylogenetic analysis of the SlMLP family unveiled their evolutionary relationships and possible functions. Furthermore, the tissue-specific expression analysis revealed that the tomato MLP members possess distinct biological functions. Crucially, multiple *cis*-regulatory elements associated with stress, hormone, light, and growth responses were identified in the promoter regions of these *SlMLP* genes, suggesting that SlMLPs are potentially involved in plant growth, development, and various stress responses. Subcellular localization demonstrated that SlMLP1, SlMLP3, and SlMLP17 are localized in the cytoplasm. In conclusion, these findings lay a foundation for further dissecting the functions of tomato *SlMLP* genes and exploring the evolutionary relationships of MLP homologs in different plants.

## 1. Introduction

Tomato is an economically important vegetable crop worldwide, having high nutritional values and health benefits for humans [1]. However, tomato production is severely affected by abiotic stresses, such as drought, flooding, high temperature, and salinity, resulting in approximately 70% yield loss depending on the severity and duration of the stress [2]. To mitigate yield losses in tomato crops, it is important to improve plant resistance to various biotic and abiotic stresses.

The major latex protein/ripening-related proteins (MLP/RRP) subfamily is a group of proteins that play a role in defense and stress responses [3,4]. The MLP homologs can be divided into three groups: the MLP, Bet v 1s, and pathogenesis-related protein class 10 (PR-10s), which are 1 of the 17 members of the pathogen-related protein (PR) family [5]. The MLP/RRP subfamily belongs to the second major family of the Bet v1 superfamily and has been identified in a variety of plant species, such as plantains (*Musa* × *paradisiaca*) [6], opium poppy [7], *Arabidopsis thaliana* [8], cucumber [9], ginseng [10], grapes [11,12], apples [13,14], kiwifruit [15], melons [16], soybean [17], cotton [18], and peanut [19]. As in the cases of Bet v 1s and PR-10s, a common structural feature of MLP proteins is the formation of a hydrophobic cavity that forms a ligand-binding site for the transport of hydrophobic compounds, such as steroids [4], long-chain fatty acids [10], and organic contaminants [20], through the phloem and xylem vessels of plants [21,22,23]. These MLP proteins have low sequence similarity but similar 3D structures [3]. They share a similar Bet v 1 fold and contain a highly conserved Gly-rich loop chain [24,25]. However, little is known about the MLP members in tomatoes.

The MLP protein was first identified in the latex of the opium poppy [26] and was subsequently reported to be associated with plant hormones [27,28] and alkaloid synthesis [29,30,31]. It is also associated with plant development [32,33,34], biotic and abiotic stresses [8,18,35], and the transport of persistent organic pollutants [20]. In wild strawberry (*Fragaria vesca*) [36], cucumber (*Cucumis sativus*) [9], kiwi (*Actinidia deliciosa*) [37], and toy pumpkin cultivars Patty Green and Gold Rush [20], the expression of *MLP* genes is closely related to fruit ripening. In addition, the overexpression of cotton *GhMLP28* in *Arabidopsis* was reported to significantly improve salt tolerance in transgenic plants [8], while the overexpression of *GhMLP28* in tobacco leaves enhances resistance to *Verticillium dahliae* infection [18,38]. Furthermore, the overexpression of *AtMLP43* in *Arabidopsis* enhances drought resistance by inducing abscisic acid (ABA) signaling [28]. In peach, the expression of *PpMLP1* is upregulated during fruit cell expansion, suggesting that *PpMLP1* may play an important role in cell and tissue expansion [27]. In addition, the expression of *MLPs* can be influenced by a variety of factors. For example, *AtMLP328* is inhibited by blue light and thus its expression is down-regulated in Arabidopsis’s recessive pigment-deficient mutants [39]. The transcript level of *MLP151* in *Panax ginseng* was increased by mannitol treatment [40]. In contrast, the expression of *MLP28* (*AT1G70830*) and *MLP43* (*AT1G70890*) was up-regulated in Arabidopsis root tips with gravity and mechanical stimulation [4,41]. The transcript levels of *VvMLP1/4/7/8* in grape were significantly increased by salicylic acid treatment, and the expression levels of *VvMLP1/2/3/6/9* were significantly increased by salt stress [12].

Although the *MLP* genes have been extensively studied in many plants, to date, the biological functions of *MLP* in tomato (*Solanum lycopersicum* L.) have not been reported. Therefore, in this study, a genome-wide search and identification of tomato *MLP* gene family members are performed by combining bioinformatics tools and molecular assays to analyze their phylogenetic relationships, gene structure, chromosome distribution and localization, conserved structural domains and patterns, and the expression of MLP family members in different organs or tissues of tomato plants, such as roots, stems, shoot apex, leaves, flowers, fruits, and seeds. To further investigate their regulatory network expression in tomato and to further understand the functions of *SlMLP* family genes, we determined their expression levels in response to cold stress, heat stress, drought stress, and salt stress by qRT-PCR. We also performed the subcellular localization of *SlMLP1*, *SlMLP3*, and *SlMLP17*. This comprehensive study of tomato *MLP* homologs reveals the potential role of these genes in plant growth, development and response to abiotic stress, and provides a basis for further studies on the functions of the tomato *MLP* gene family.

## 2. Results

### 2.1. Identification and Characterization of the MLP Family Genes in Tomato

Thirty-four SlMLP gene sequences with complete Bet v1 allergen domains were identified in the tomato genome. We then named these genes SlMLP1–SlMLP34 according to their order of arrangement on the chromosome. The physicochemical properties of these genes are shown in Table 1. The coding sequence size (CDS) of these 34 genes ranges from 441 to 891 nucleotides and the protein length ranges from 146 to 296 amino acids. Among these genes, SlMLP6 has the lowest relative molecular mass of 16.57 kDa, while SlMLP10 has the highest relative molecular mass of 34.40 kDa. The isoelectric points of the 34 MLP proteins range from 4.78 to 8.73. All members are acidic, except SlMLP32, which is basic. The GRAVY values of all 34 MLP proteins are less than zero, indicating that these proteins are hydrophilic. The instability coefficients range from 16.91 to 51.29, with all SlMLPs, except SlMLP15 and SlMLP16, having low instability indices (<40), indicating that 82.3% of SlMLPs are stable at the theoretical level. Subcellular localization predictions suggest that most SlMLP proteins may target the cytoplasm, except SlMLP11 and SlMLP31, which may be localized in the nucleus and chloroplasts, respectively (Table 1).

### 2.2. Phylogenetic Tree Analysis of SlMLP Genes

To explore the evolutionary relationships of MLP homologous genes, 132 MLP proteins were obtained from four plant species, including tomato, Arabidopsis, apple, and cucumber, and a phylogenetic tree was constructed by the neighbor-joining (NJ) method (Figure 1). Based on the classification of MLP families in Arabidopsis, apple, and cucumber [12,14,42], our phylogenetic tree divided these 132 MLP homologs into three groups. A total of 23 *SlMLP*, 6 *CsMLP*, 6 *MdMLP*, and 10 *AtMLP* were clustered within Group I; 3 *Mdmlp*, 19 *CsMLP*, and 14 *AtMLP* were clustered within Group II; and finally, 11 *SlMLP*, 27 *MdMLP*, 12 *CsMLP*, and 1 *AtMLP* were clustered within Group III. Furthermore, most tomato *SlMLP* genes have multiple homologous members in apple and fewer homologous members in other species, suggesting evolutionary independence among different species. Interestingly, *SlMLP3* and *SlMLP11* have a separate evolutionary branch. Similarly, *SlMLP32* and *SlMLP18* also have a separate evolutionary branch. Overall, the phylogenetic tree analysis revealed a highly conserved amino acid sequence, suggesting a strong evolutionary relationship between each member of the tomato MLP family.

### 2.3. Gene Structure, Conserved Motif, and Domain Analysis of SlMLPs

We also established a phylogenetic tree of SlMLP, followed by gene structure information and gene motif analysis (Figure 2). The results of the gene structure analysis show that the number of exons in all *SlMLP* genes ranged from two to three, with the majority (32 members) containing two exons and two members (*SlMLP10* and *SlMLP11*) having three exons. Most of the coding sequence is disrupted by introns. Notably, unlike the other *SlMLP* genes, the *SlMLP10* gene is the longest (4382 bp), suggesting that a different pattern of evolution has occurred in this gene.

To further investigate the diversity of SlMLPs, ten different conserved motifs were identified in the SlMLP sequence by MEME motif analysis (Figure 2b; Appendix A). These conserved motifs range in length from 11 to 57 amino acids and are widely distributed among all SlMLP proteins. Of these, the SlMLP32 protein contains only one motif (motif 3), while the SlMLP10 protein has all 10 motifs. Notably, members that are phylogenetically closely related have a similar motif composition. According to the results of motif analysis, motif 4, motif 7, motif 8, and motif 10 are specific to members of Group III and are only present in some members. Motif 3 was detected in all SlMLP proteins, while motifs 1 and 2 were detected in 24 members and motifs 5 and 6 were detected in 23 members, indicating that most motifs are conserved in the SlMLP family. The similarity in characteristic patterns of SlMLP proteins may reflect a functional similarity, while the functional differences in SlMLP genes may be due to the different distributions of conserved motifs.

Furthermore, conserved structural domain analysis revealed that the structural domains of all members of SlMLPs are highly conserved, and they contain three structural domains, the SRPBCC superfamily or Bet v 1 or Bet v1-like. The members of phylogenetically closely related SlMLPs have a similar structural domain composition, implying that their primary functions are likely to be the same (Appendix A).

### 2.4. Chromosomal Location and Synteny Analysis

To determine the chromosomal distribution of *SlMLP* genes, we localized them to the tomato genome (ITAG4.0). The results show that 34 *SlMLP* genes are randomly distributed on 8 of the 12 tomato chromosomes, of which chr04 and chr09 contain 11 members; chr05 contains 4 genes; chr03, chr07, and chr10 contain 2 genes; and chr08 and chr12 contain only 1 gene each (Figure 3). Chr04 and chr09 contain the largest number of *SlMLP* members (11/34, 32.4%).

To further understand the evolutionary clues of *SlMLP* members, we conducted a co-linear analysis of tomato and other plant species (Arabidopsis, apple, and cucumber). The results show that a total of two *SlMLP* members are involved in co-linear relationships. One *SlMLP* member is homologous to the Arabidopsis gene, one *SlMLP* member is homologous to the cucumber gene, and two *SlMLP* members are homologous to the apple gene. Of these, only one *SlMLP* member (*SlMLP14*) showed paired homology to genes from *A. thaliana*, cucumber, and rice. Overall, there are one, one, and three co-linear gene pairs between tomato and Arabidopsis, tomato and apple, and tomato and cucumber, respectively (Figure 4). This indicates that the *SlMLP* genes of tomato, Arabidopsis, apple, and cucumber are poorly diversified evolutionarily trajectories and the number of homologous genes is relatively high. Taken together, the results suggest that the *SlMLP* family genes may have similar functions to the MLPs of other species.

### 2.5. Cis-Element Analysis of SlMLP Promoters

Given the importance of cis-acting elements in the regulation of gene expression, it is necessary to predict and analyze the cis-acting elements in the promoter region of the SlMLP genes before exploring their expression patterns. For this purpose, the 2 kb promoter sequences of the SlMLP genes were obtained and submitted to the PlantCare database. The PlantCare-based analysis showed that all SlMLP genes have multiple regulatory elements. In addition to the conventional core elements detected in all promoters of SMLPs and some core elements of unknown function, we identified 32 major cis-acting regulatory elements of SlMLPs. Among them, SlMLP31 has the highest number of regulatory elements, while SlMLP22 has the lowest number of regulatory elements (Figure 5f). They can be classified into four categories, namely plant growth and development, phytohormone response, stress response, and light response (Figure 5e). Among all 34 SlMLP promoter sequences, hormone-related elements accounted for the largest proportion (32.4%), and light-responsive cis-regulatory elements were the second largest category (31.4%). Other categories, such as stress-related elements, accounted for 28.5% of the total and growth and development-related elements accounted for 7.7% of the total. The plant growth and development group includes CAT-box, CCGTCC-box, Circadian, Gcn4_Motif, HD-Zip 1, O2-Site, and RY-Element, which are responsible for phloem expression, phloem activation, circadian regulation, endosperm expression, fenestrated chloroplast differentiation, maize alcoholic protein metabolism, and seed development, respectively. Among them, circadian elements are most abundant in the SlMLP promoters (Figure 5a). The phytohormone response group includes elements responsive to growth hormone (AuxRR-core and TGA-Element), gibberellin (Gare-Motif, P-box and TATC-box), methyl jasmonate (CGTCA-Motif and TGACG-Motif), abscisic acid (Abre), ethylene (ERE), and salicylic acid (TCA-Element), with the largest number of ERE elements (Figure 5b). In the stress response group, there are many elements involved in anaerobic induction (ARE and GC-Motif), dehydration (MBS), hypothermia (LTR), stress (STRE), defense (TC-rich repeats), and trauma (WRE3 and WUN-Motif), with the largest number of STRE elements (Figure 5c). Light-responsive cis-regulatory elements include G-box, MRE, 4cl-CMA2b, Box 4, GT1-motif, Sp1, and ACE, with the largest number of Box 4 elements (Figure 5d). In addition, in most SlMLP members, each SlMLP gene has different types and numbers of cis-elements, and the proportion of elements in the phytohormone response and light response categories is higher than in the plant growth and development group or stress response group (Figure 5f). Taken together, our analysis suggests that SlMLP genes may be involved in various biological processes in tomato.

### 2.6. Tissue-Specific Expression Patterns of SlMLP Genes

To investigate the potential functions of *SlMLP* genes, their expression patterns in tomato root, stem, stem tip, leaf, flower, fruit, and seed tissues were analyzed using qRT-PCR. Then, the qRT-PCR data were used to construct a heat map. As shown in Figure 6, SlMLP members in the same branch showed similar expression profiles with minimal differences. For example, *SlMLP1*, *SlMLP2*, *SlMLP23*, *SlMLP24*, *SlMLP28*, and *SlMLP31* were mainly expressed in seeds, whereas *SlMLP23*, *SlMLP24*, and *SlMLP31* were also expressed at higher levels in roots. Meanwhile, *SlMLP4*, *SlMLP5*, *SlMLP10*, *SlMLP21*, *SlMLP25*, *SlMLP26*, and *SlMLP27* were mainly expressed in the roots, among which *SlMLP10*, *SlMLP25*, *SlMLP26*, and *SlMLP27* were also expressed at higher levels in the stems. *SlMLP6*, *SlMLP9*, *SlMLP17*, *SlMLP18*, *SlMLP22*, and *SlMLP34* were mainly expressed in leaves; *SlMLP16* and *SlMLP20* were mainly expressed in stem tips; and *SlMLP3*, *SlMLP8*, *SlMLP11*, *SlMLP12*, *SlMLP13*, *SlMLP15*, *SlMLP19*, and *SlMLP33* were mainly expressed in flowers. *SlMLP7*, *SlMLP14*, *SlMLP29*, *SlMLP30*, and *SlMLP32* were mainly expressed in fruit. Surprisingly, most of the *SlMLP* members had a very low expression in the stem tip (Figure 6). These results suggest that *SlMLP* genes may play different roles in the regulation of growth and development of tomato plants and that there are both functional similarities and functional differences among these members.

### 2.7. Expression Profiles of SlMLPs in Response to Abiotic Stresses

In the present study, cis-elements associated with stress response were identified in the promoter of tomato *SlMLP* genes (Figure 5). Therefore, we randomly selected 10 and 6 members from group Ⅰ and group Ⅲ, respectively, for further analysis of the expression of *SlMLPs* in tomato leaves under different abiotic stresses, such as heat, cold, salt, and drought. Overall, these 16 *SlMLP* members showed up- or down-regulation in response to different stresses to different degrees (Figure 7, Appendix A). Under high temperature (42 °C) treatment, four of these genes were up-regulated and six were down-regulated in expression (Figure 7, orange bars). The transcript levels of *SlMLP1*, *SlMLP4*, *SlMLP7*, and *SlMLP17* were elevated at all time points after treatment. *SlMLP11* was induced at some time points and decreased to control levels at 24 h. In addition, *SlMLP2*, *SlMLP3*, *SlMLP9*, *SlMLP10*, *SlMLP19*, and *SlMLP27* showed similar expression patterns and were repressed at all time points of treatment (Figure 7, orange bars), and in addition, the other six *SlMLP* members showed inconsistent expression profiles under high temperature treatment (Appendix A, orange bars).

Under low temperature treatment (4 °C), four *SlMLP* genes were induced, whereas 11 members were repressed (Figure 7, blue bars; Appendix A, blue bars). Among them, *SlMLP4* and *SlMLP17* showed significantly higher transcript levels than the control (0 h) throughout the experiment (Figure 7, blue bars). The expression of *SlMLP7* was elevated at certain time points, but eventually decreased to normal levels (Appendix A, blue bars). In contrast, *SlMLP2*, *SlMLP3*, *SlMLP10*, *SlMLP19*, *SlMLP27*, and *SlMLP30* mRNA levels were significantly lower at different time points after treatment (Figure 7, blue bars). In addition, *SlMLP11* exhibited a disordered expression pattern (Appendix A, blue bars).

In the case of simulated salt stress with NaCl, four *SlMLP* genes were induced, while five members were repressed. Among them, *SlMLP1*, *SlMLP4*, *SlMLP7*, and *SlMLP17* showed significantly higher transcript levels than the control (0 h) throughout the experimental period. In contrast, *SlMLP3*, *SlMLP8*, *SlMLP27*, *SlMLP29*, and *SlMLP30* mRNA levels were significantly lower at different time points after treatment (Figure 7, brown bars). In addition, the expressions of *SlMLP9*, *SlMLP18*, and *SlMLP33* showed a trend of increasing followed by decreasing, while *SlMLP2*, *SlMLP10*, *SlMLP11*, and *SlMLP19* showed a disordered expression pattern (Appendix A, brown bars).

The numbers of positively and negatively affected *SlMLP* genes under simulated drought stress with polyethylene glycol 6000 were 7 and 4, respectively (Figure 7, green bars; Appendix A, green bars). For example, the expression of *SlMLP1*, *SlMLP4*, *SlMLP7*, and *SlMLP17* increased at different time points, but the transcript abundance of *SlMLP9*, *SlMLP10*, and *SlMLP30* increased at 6 h or 12 h and remained high at 12 h. In contrast, the expression of *SlMLP3*, *SlMLP8*, *SlMLP19*, and *SlMLP33* decreased at different time points and the repression persisted until 12 h (Figure 7, green bars). In addition, *SlMLP2*, *SlMLP11*, *SlMLP18*, *SlMLP27*, and *SlMLP29* exhibited disordered expression patterns under drought stress (Appendix A, green bars).

### 2.8. Subcellular Localization of SlMLP Proteins

Most of the SlMLP proteins were predicted to be localized in the cytoplasm (Table 1). To verify these results, three SlMLP members, SlMLP1, SlMLP3, and SlMLP17, were selected and their expression trends were similar across treatments in response to the four abiotic stress treatments. Transient expression in the leaf epidermis of tobacco leaves was analyzed using an *Agrobacterium*-mediated assay and the fluorescence signal was observed by laser confocal microscopy. SlMLP1, SlMLP3, and SlMLP17 were localized in the nucleus, membrane, and cytoplasm, but the signals in the cytoplasm were weak (Figure 8), which is similar to our prediction (Table 1). As a control, the 35S-GFP signal was detected throughout the cells (Figure 8).

### 2.9. Analysis of the SlMLP Gene Expression Network

To explore the protein interaction network of the tomato MLP family, a protein interaction analysis was performed using the string database with *SlMLPs* as the query sequence and tomato as the target comparison species. The predicted results of the SlMLPs interaction network relationships show that SlMLP33 interacts with many genes, including genes with a proven function (loxC) in addition to six genes with an unknown function of the gene (Figure 9). LoxC has been shown to be involved in many different aspects of plant physiology, including growth and development, pest resistance, senescence, or response to injury. Additionally, *SlMLP29* and *SlMLP30* interact with profin genes. Profin gene has been shown to bind to actin and affect the structure of the cytoskeleton. Therefore, we hypothesize that *SlMLP33*, *SlMLP29*, and *SlMLP30* play important roles in the regulation of abiotic and biotic stress resistance in tomato.

## 3. Discussion

Bet v1 fold has been reported to be an ancient and versatile scaffold for binding large hydrophobic ligands [3]. Notably, plant-specific MLPs belong to the Bet v1 protein family [18]. MLP genes play important roles in plant growth and development [22], as well as responses to biotic and abiotic stresses [8,13,18]. Plant MLPs are also involved in the transport of hydrophobic compounds [5]. In the present study, the availability of the tomato genome sequence provides us an opportunity to study the characteristics of the MLP family in tomato [43,44].

### 3.1. Conservation and Evolution of the MLP Family

Since the first MLP protein was isolated from the latex of the poppy (*Papaver somniferum*) [6], a number of MLP proteins have been found in many dicotyledons, monocotyledons, and conifers. In recent years, MLP genes have been extensively identified and studied in many plants, such as *Brassica rapa* [45], *Cucurbita pepo* [46], grape [12], apple [14], and cucumber [42]. However, to date, limited attention has been paid to members of the tomato MLP gene family. In the present study, we identified 34 MLP genes in tomato by genome-wide identification (Table 1). Meme motif analysis showed that the MLP protein structure is widely conserved in tomato. Based on protein structure analysis, the SlMLP protein has a conserved SRPBCC superfamily or Bet v1 or Bet v1-like structural domain (Appendix A), which is typical for the MLP family. Only one Bet v1 structural domain was found in the MLP proteins of cucumber [42], apple [14], and oilseed rape [45], while up to four Bet v1 structural domains were found in zucchini [46]. These results suggest that MLP proteins are evolutionarily conserved and the most diverse among species.

Notably, in our structural domain analysis, we found that these 34 MLPs contain only one Bet v 1 structural domain, or only one SRPBCC superfamily, or only one Bet v1-like structural domain. These three functional structural domains account for a significant proportion of the total protein length. It is clear that most of the coding regions of SlMLPs encode either a Bet v 1 structural domain or an SRPBCC superfamily, or a Bet v1-like structural domain (Figure 2; Appendix A).

In previous studies, it was found that, in monocotyledons, such as Zea mays, there are only three MLPs; *Brachypodium distachyon* has only two MLPs; and *Setaria italica* has only one MLP. These numbers were much lower than those of dicotyledons, such as *Populus tremula* (10 MLPs), *Fragaria vesca* (13 MLPs), *Arabidopsis thaliana* (25 MLPs), and *B. rapa* (31 MLP) [45]. We hypothesize that the MLP subfamily might have diverged after the monocot–dicot division during plant evolution; this process appears to have led to the disappearance of MLPs in monocots and the development of MLPs in dicots.

The analysis of the 34 SlMLP amino acid motifs identified five relatively conserved motifs (motif 1, motif 2, motif 3, motif 5, and motif 6) (Figure 2). There are many glycines, lysines, and glutamates as well as several prolines. Notably, glycine protects the photosynthesis mechanism of plants from damage and works efficiently under drought conditions [47]. Lysine is known to be associated with photosynthesis in plants [48]. Glutamate is a signal used by plants in response to wounds. When the plant senses a localized signal, it systemically transmits this information to the entire plant, which rapidly activates a defense response in the intact part of the plant [49]. However, proline causes osmotic pressure to decrease under drought stress and plays an important role in plant development [50]. Therefore, we hypothesized that these five conserved motifs may be important functional regions of the Bet v 1 domain.

### 3.2. Conservation and Evolution of the MLP Family

We predicted the subcellular localization of SlMLPs (Table 1). Interestingly, most SlMLPs scored the highest in the cytoplasm. We selected three members to assay the subcellular localization for validation, and the experimental results show that SlMLP1, SlMLP3, and SlMLP17 were all localized in the cytoplasm (Figure 8). This result is consistent with a previous study, where MLPs were found to be concentrated in the cytoplasm of alkaloid vesicles when isolated from the poppy [51]. This evidence further supports the localization of the identified SlMLP family genes.

### 3.3. Cis-Element Analysis of SlMLP Promoters

Cis-acting promoter elements are involved in the regulation of gene expression through interactions between promoter-binding sites and transcription factors [52]. The results indicate that 34 SlMLP genes contain multiple functional elements, such as STRE, Box 4, ERE, ABRE, and WUN-motif elements (Appendix A). In addition, ERE elements appeared in 28 SlMLP (82.4%) genes, which are thought to be targets of ABA or ethylene signaling and play important roles in ethylene and ABA regulation in plants [53,54]. These results are consistent with the analyses of cis-acting elements in crops such as apple [14] and suggest that MLP genes may play an important role in regulating plant growth and adaptation to environmental stress [55,56,57].

### 3.4. Potential Functions of Tomato SlMLP Genes in the Regulation of Plant Growth and Development

MLP is considered to be an important regulator of a range of plant developmental processes [32,33,34]. In the present study, we found that tomato SlMLP members exhibited different expression profiles in different tissues (Figure 6). Taken together, 10 members of the SlMLP family are highly expressed in roots, four in stems, six in leaves, eight in flowers, five in fruits, and six in seeds, suggesting that the SlMLP family is similar to MLP homologs in other plants and may be involved in multiple aspects of tomato plant growth and development, and some of these members may have similar expression patterns due to their redundant functions due to their similar expression patterns.

### 3.5. SlMLP Is Involved in the Response to Abiotic Stresses

Previous studies have reported the response of MLPs to various abiotic stresses, for example, VvMLP1/2/3/6/9 in grapes, showed significantly increased transcript levels after salt stress treatment [12], and MdMLP5/13 in apple also showed a significantly up-regulated expression after salt stress treatment [14]. The expression levels of MdMLP2/6/7/9/11 were up-regulated after drought treatment [14]. SlMLP14 is homozygous with AtMLP423 (Figure 4). T-DNA knockout insertion mutant mlp423 shows that MLP423 is essential for normal development in Arabidopsis and mlp423 knockout causes slight changes in leaf curvature in Arabidopsis [33]. However, AAAP fungal infection significantly reduced the expression of a homolog of AtMLP423, MdMLP22, in apple [14]. In our work, we found that tomato SlMLP genes also responded to various abiotic stresses (Figure 7). Overall, the low temperature treatment induced 4 SlMLP members and repressed 11 members. High-temperature and drought treatments increased the transcript levels of five and seven SlMLP genes, respectively, and decreased the transcript levels of seven and seven genes, respectively. Salt treatment induced 4 SlMLP members and repressed 10 members. Thus, the number of up-regulated SlMLP family members was lower than the number of down-regulated members under low temperature, high temperature, drought stress and salt stress, suggesting that the SlMLP family plays a minor role in the tolerance of plants to these four stresses. In addition, a Venn diagram showed that four SlMLP genes were involved in three stresses (Appendix A), with SlMLP1 and SlMLP4 being significantly up-regulated after four treatments (Figure 7), suggesting that they may be key regulatory genes for stress tolerance in tomato. These observations suggest that the MLP family is extensively involved in plant responses to different abiotic stresses.

## 4. Materials and Methods

### 4.1. Plant Materials and Growth Conditions

The tomato (*S. lycopersicum* L.) cultivar Ailsa Craig (AC) was used in this study. Seeds were sown in 50-hole cavity trays filled with nutrient soil (charcoal:vermiculite = 1:1). They were then cultured in a growth chamber with a photoperiod of 16 h/8 h (light/dark) and a day/night temperature of 25 °C/18 °C. For the tissue-specific expression analysis, 4-week-old seedlings were transplanted into the greenhouse. At the adult stage, roots, stems, stem tips, and leaves of plants, fully opened flowers at anthesis, fruits 7 days after flowering (anthesis), and seeds of fruits in the mature red stage were collected and immediately frozen in liquid nitrogen. Each tissue was prepared with three independent biological samples of 100 mg each for RNA extraction.

To perform expression analysis under different abiotic stresses, plants were always cultured in growth chambers. Different stress treatments were applied to four-week-old tomato plants. To impose cold and heat stress, plants were incubated at 4 °C and 42 °C, respectively. Leaves were collected at different time points after treatment (0 h, 1 h, 3 h, 6 h, 12 h, and 24 h). For salt stress treatment, plants were watered with 200 mM NaCl (50 mL per plant) and sampled after 0 h, 1 h, 3 h, 6 h, 12 h, and 24 h. For drought induction, plants were watered with 20% polyethylene glycol 6000 (polyethylene glycol) (50 mL per plant) and sampled after 0 h, 3 h, 6 h, 9 h, and 12 h. Tomato plants were grown under the same substrate conditions (both composition and weight) with three biological replicates for each treatment and each treatment group consisted of three tomato seedlings.

### 4.2. Identification of SlMLP Genes in Tomato

To identify all members of the tomato MLP family, a Hidden Markov Model (HMM) file of the MLP structural domain (PF00407) was downloaded from the Protein Family (PFAM) database and queried against the HMMER 3.0 expected value (E-value) of 1 × 10^−5^ [58] for the Solanaceae Genomics Network (SGN; http://www.solgenomics.net (accessed on 10 March 2023)) to search for SlMLP genes. Subsequently, using the SMART database (http://smart.embl-heidelberg.de/ (accessed on 12 March 2023)) [59] and the NCBI Conserved Domains Database (http://www.ncbi.nlm.nih/gov/Structure/cdd/wrpsb.cgi (accessed on 15 March 2023)), all putative tomato MLP genes were identified. The protein sequences of SlMLPs were analyzed using the Prosite ExPASy server (http://web.expasy.org/protparam/ (accessed on 16 March 2023)) to predict their physical and chemical properties. These properties include molecular weight (MW), protein length based on amino acid number (AA), theoretical isoelectric point (PI), total average hydrophilicity (gravy), and instability index (instability index). In addition, subcellular localization predictions were performed using the online software WoLF PSORT (https://www.genscript.com/wolf-psort.html (accessed on 20 March 2023)). Other features, such as gene localization and coding sequence length (bp), were obtained from the Solanaceae genome database.

### 4.3. Phylogenetic Analysis

MLP direct homologs from a range of species were used for phylogenetic analysis. These protein sequences were obtained from the databases of the Solanaceae Genome Network, the Arabidopsis Information Resource (https://www.arabidopsis.org/ (accessed on 1 April 2023)), the Apple Protein Sequence Database (https://iris.angers.inra.fr/gddh13/ (accessed on 3 April 2023)), and the Cucumber Genome Database (http://cucurbitgenomics.org/ (accessed on 5 April 2023)). A total of 1000 bootstrap replicates were performed using MEGA 7.0 software, which was further used for multiple sequence comparisons and phylogenetic analysis by the neighbor-joining method. Afterward, the phylogenetic tree was visualized and annotated using the iTOL tool (https://itol.embl.de/ (accessed on 10 April 2023)).

### 4.4. Analysis of Motifs, Gene Structures, and Conserved Domains

The positions of exons, introns, and untranslated regions of each SlMLP gene were obtained from the Solanaceae genome. Conserved motifs in SlMLPs were identified using the MEME server (https://meme-suite.org/ (accessed on 20 April 2023))) with the following parameters: maximum motif number of 10, minimum motif width of 6, and maximum motif width of 100 [60]. We used the NCBI Conserved Structural Domain Database (CDD) for structural domain analysis to determine the type of structural domain and the location of all SlMLP sequences. The exon/intron structure of the SlMLP genes, as well as the conserved motifs and structural domains of the SlMLP proteins, were visualized using TBTools software [61].

### 4.5. Chromosome Localization and Synteny Analysis

Based on chromosome length and gene location information obtained from the tomato genome annotation file (ITAG 4.0), the localization of the SlMLP genes on the corresponding chromosome was determined by TBtools. Homologous gene pairs and synonymous relationships for the MLP gene family in tomato were determined using the Multiple Covariance Scanning Toolkit (MCScanX)(1.0) software [62] and default parameters. MLP gene duplication and homologous genetic relationships of MLP genes in tomato and other species (Arabidopsis, cucumber, and apple) were visualized using TBtools.

### 4.6. Analysis of the MLP Gene Promoter in Tomato

The 2 kb promoter sequence (before the start codon) of each SlMLP gene was extracted using TBTools and submitted to the online database PlantCare (http://www.bioinformatics.psb.ugent (accessed on 26 April 2023); BE/WebTools/PlantCare/html/) to predict the cis-elements. The predicted cis-regulatory elements were classified according to their regulatory function, and cis-regulatory elements associated with growth and development, light response, hormones, and environmental stress were visualized through TBTools.

### 4.7. RNA Extraction and qRT-PCR Analysis

The total RNA was extracted from the above-mentioned samples (Section 4.1) using an RNA extraction kit (TaKaRa). The PrimeScript RT reagent kit (TaKaRa) was used for cDNA synthesis. The qRT-PCR assay was conducted with the SYBR Premix Ex Taq kit (TaKaRa) using cDNA as the template, and the Applied Biosystems 7500 Real-Time PCR System (Applied Biosystems, Foster City, CA, USA) was used to conduct this assay. The EF-1α gene (Gene ID: Solyc03g119290) was used as the internal control. The relative expression of the target genes was calculated through the 2^−ΔΔCt^ method [63]. Three independent biological samples were used for each qRT-PCR analysis, and three technical replicates for each cDNA sample. For tissue-specific expression, the qRT-PCR data were log2-normalized and visualized as a heat map using TBtools. The gene-specific primers are listed in Appendix A.

### 4.8. Subcellular Localization

The selected *SlMLP* gene was cloned into the CAM-EGFP vector. The constructed vector was transformed into Agrobacterium tumefaciens GV3101 and then infiltrated into the surrounding tobacco (*Nicotiana benthamiana*) leaves and incubated in the dark at 25 °C for 48 h. Afterwards, the infiltrated leaves were placed on slides and the fluorescence signal was observed using an Olympus BX53 fluorescence microscope. The analysis was performed only on the leaf epidermis. The primers used in the study are listed in Appendix A.

### 4.9. Interaction Network Prediction

The amino acid sequences of 16 SlMLPs were submitted to the STRING website (https://string-db.org/ (accessed on 20 April 2023)) for the prediction and construction of interaction networks. A minimum score of 0.400 was set for interactions. Active interaction data were obtained from curated databases, experimental assays, gene neighboring, gene fusion, gene co-occurrence, text mining, co-expression, and protein homology.

## 5. Conclusions

In the present study, we identified 34 *SlMLP* genes in tomato. The phylogenetic analysis classified them into three subfamilies. SlMLP proteins contain conserved SRPBCC structural domains or Bet v 1 structural domains or Bet v1-like structural domains, which are typical features of the MLP family. The analysis of protein motifs showed that most MLPs in tomato are relatively conserved. A total of 34 *MLP* genes are distributed on each of the eight chromosomes of tomato. Based on the cis-element prediction and expression analysis, we hypothesized that *SlMLP* genes may be involved in the regulation of various aspects of plant growth and development, as well as in the response of plants to abiotic stresses. Thus, our work provides valuable information for the further understanding of the precise functions of *SlMLP* genes.

## Figures and Tables

**Figure 1 ijms-24-15005-f001:**
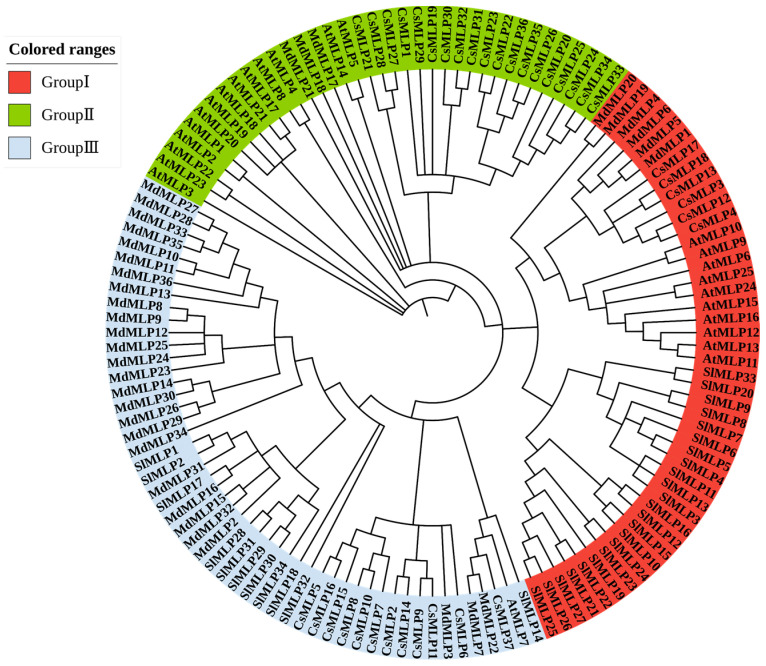
Phylogenetic analysis of MLP direct homologs in different plants. The MLP family is divided into three groups, represented by different colors. At, Arabidopsis; Sl, tomato; Md, apple; Cs, cucumber.

**Figure 2 ijms-24-15005-f002:**
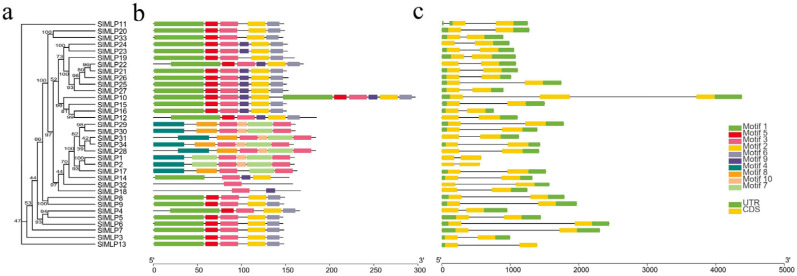
Phylogenetic analysis, gene structure, and conserved motifs of tomato MLP genes. (**a**) Construction of NJ tree consisting of 34 SlMLP protein sequences. (**b**) Distribution of conserved motifs in MLP proteins. (**c**) Exon/intron structure of the SlMLP gene. Different colored boxes represent different themes. The length of the motifs can be estimated using the scale at the bottom.

**Figure 3 ijms-24-15005-f003:**
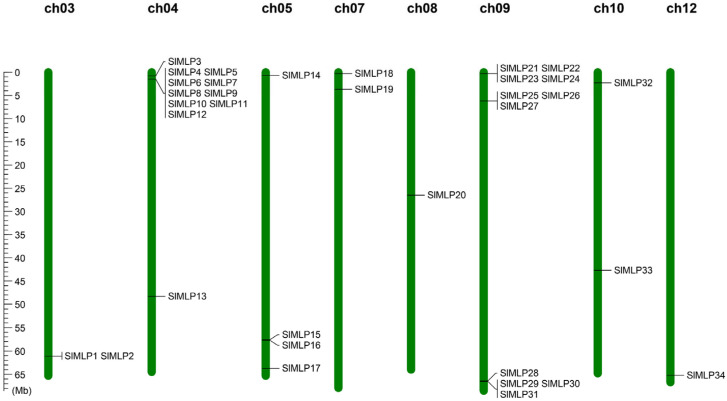
Distribution of the 34 SlMLP genes on the 8 chromosomes. Vertical bars represent chromosomes, and chromosome numbers are at the top of each chromosome. The scale on the left represents the chromosome length (Mb).

**Figure 4 ijms-24-15005-f004:**
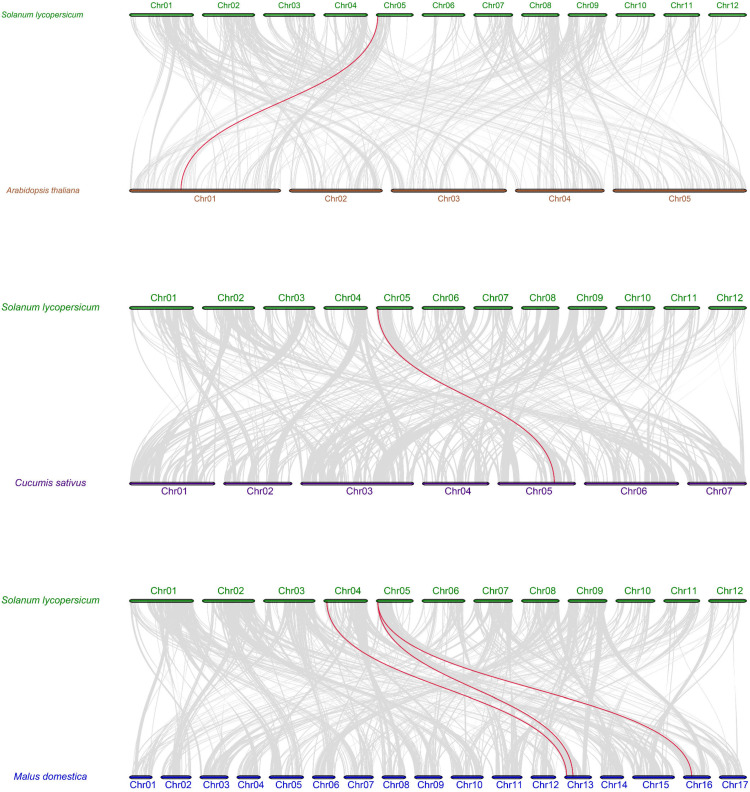
Syntenic analysis of tomato and three other plant MLP genes. All homozygous blocks between the two genomes are indicated by gray lines in the background, and homozygous MLP gene pairs are marked by red lines. The numbers indicate the order of the chromosomes.

**Figure 5 ijms-24-15005-f005:**
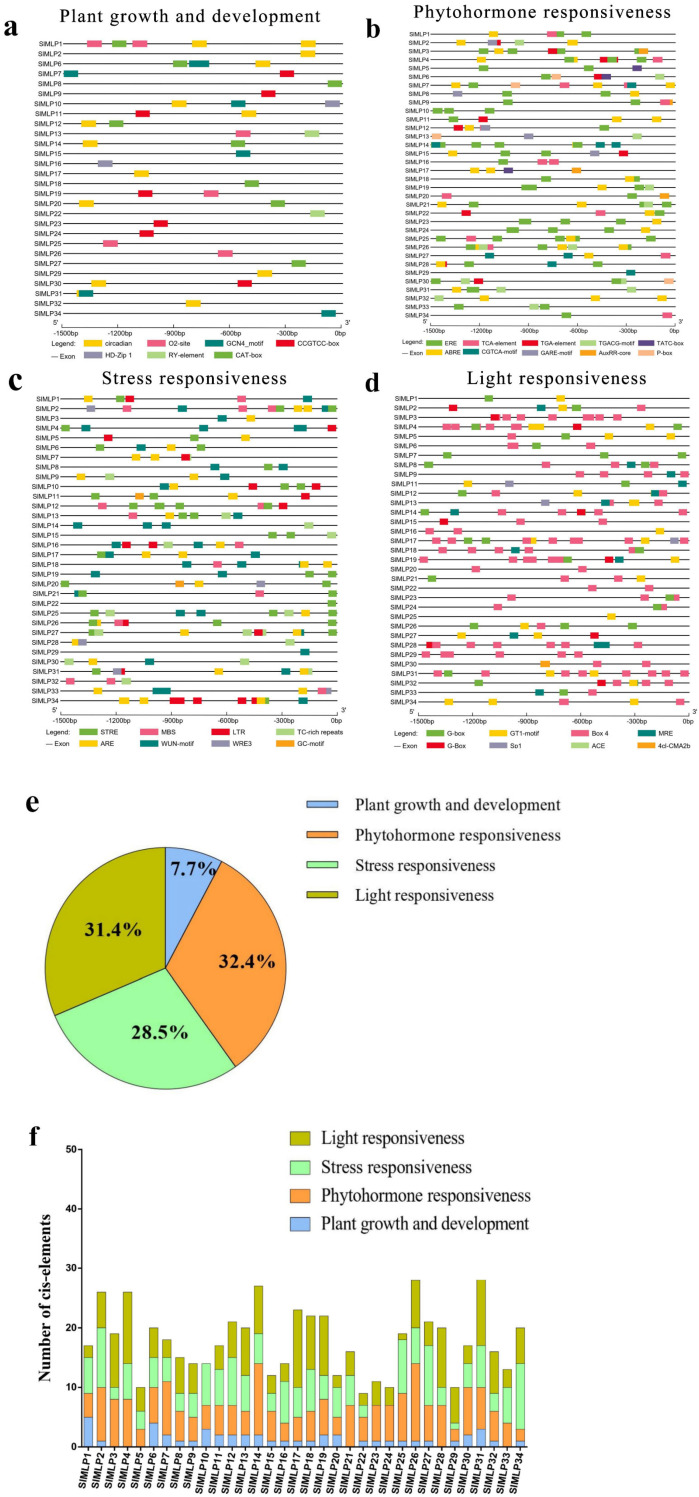
Cis-element analysis of the SlMLP promoters. (**a**–**d**) SlMLP cis-elements of promoters are classified into different groups. The elements are indicated by differently colored boxes. (**e**) The proportion of cis-regulatory elements in the promoter region of the SlMLP gene. (**f**) Number of cis-elements in four groups for each SlMLP promoter.

**Figure 6 ijms-24-15005-f006:**
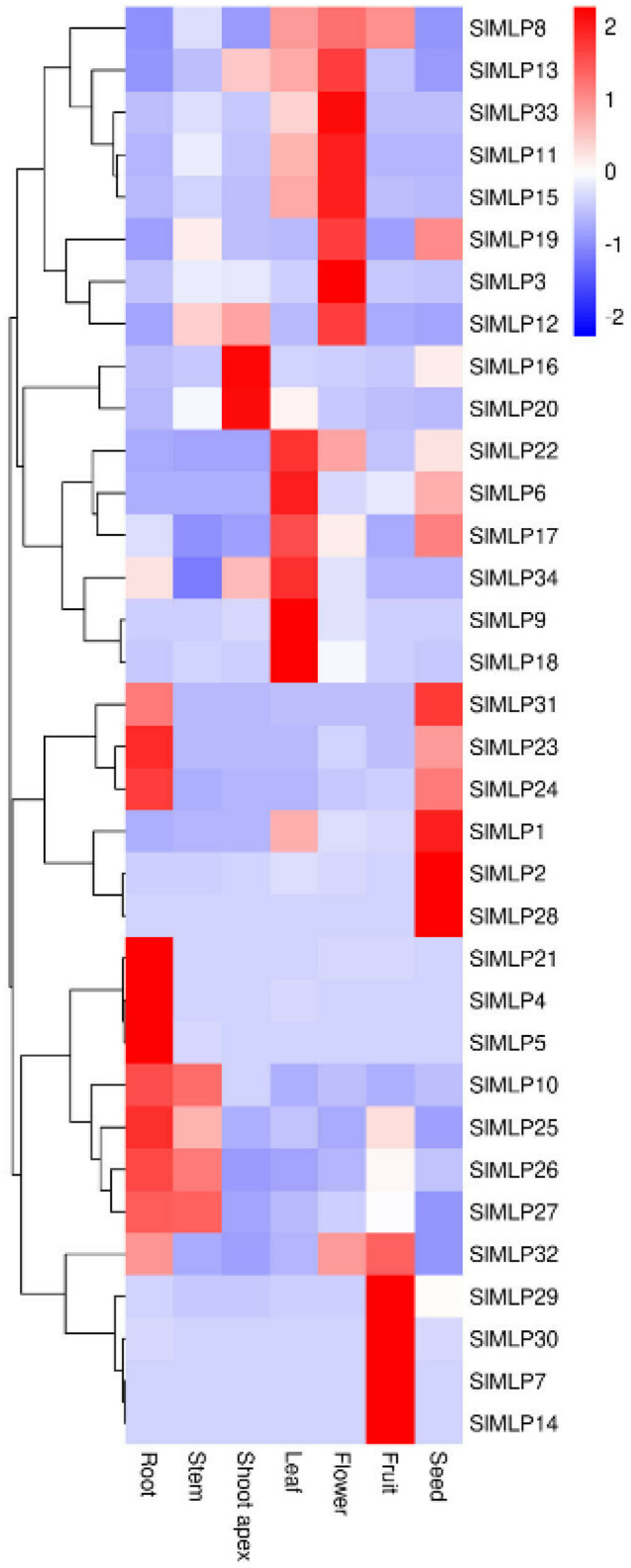
Expression patterns of SlMLPs in various tomato tissues. The qRT-PCR data were log2-normalized to construct the heat map using TBtools (1.0) software. The roots, stems, shoot apices, and leaves from plants at the 4 true-leaf stage, fully opened flowers, fruits at 7 DPA (days post-anthesis), and seeds from fruits at the mature red stage were used for this analysis.

**Figure 7 ijms-24-15005-f007:**
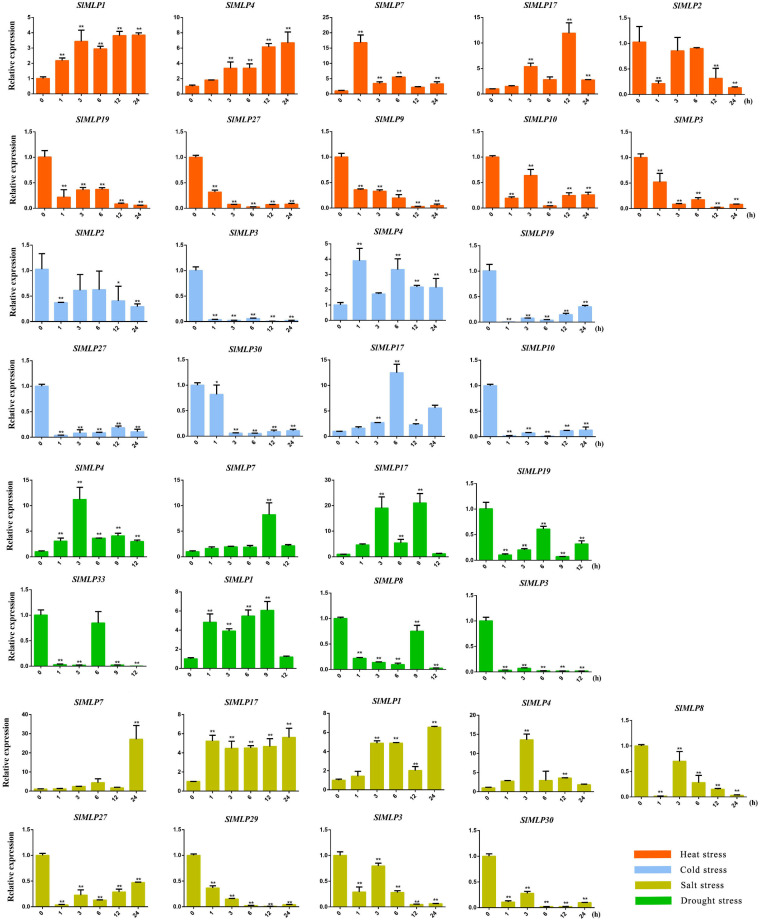
qRT-PCR analyses of SlMLP genes under different abiotic stresses. The relative expression levels of SMLPs under heat (42 °C), cold (4 °C), salt (NaCl), and drought (PEG-6000) are marked in orange, blue, brown, and green colors, respectively. Each value is the mean ± SD of three biological replicates and vertical bars indicate standard deviation. Asterisks represent significant differences in gene expression between abiotic stress treatments (different time points) and the control (0 h) by Student’s *t*-tests (* *p* < 0.05, ** *p* < 0.01).

**Figure 8 ijms-24-15005-f008:**
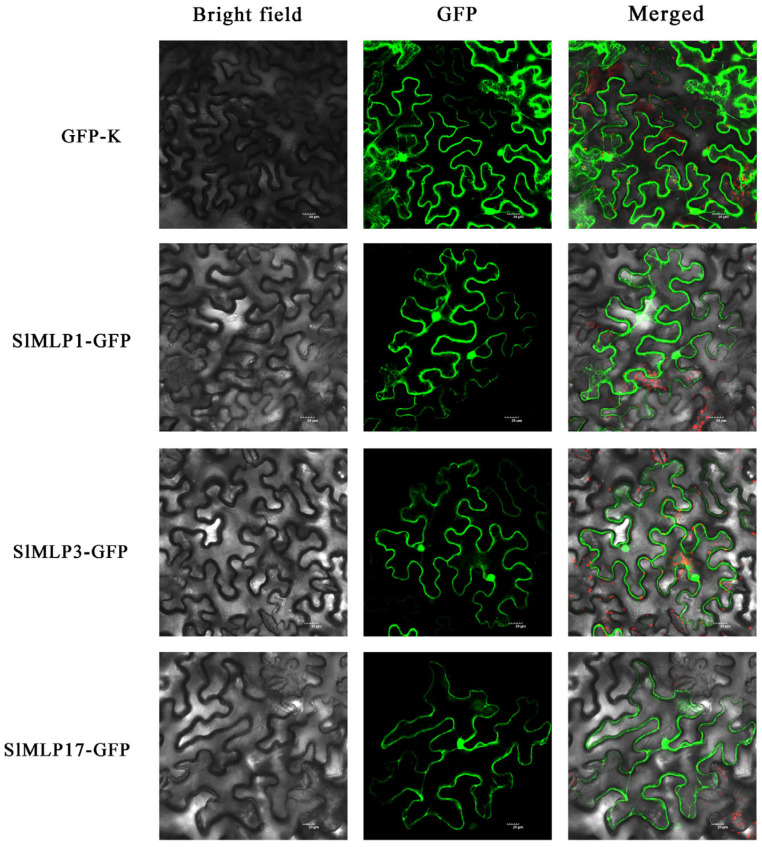
Subcellular localization of SlMLP proteins. GFP signals showing the subcellular localization of the selected SlMLP proteins. Bars = 20 μm.

**Figure 9 ijms-24-15005-f009:**
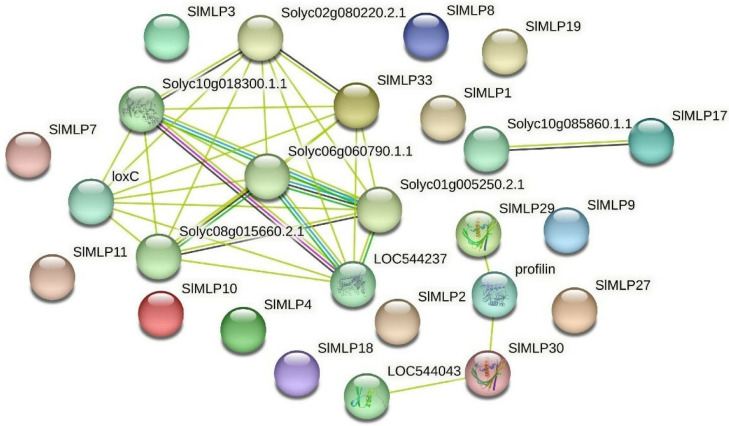
Predicted interaction network of SlMLP proteins. The different data sources of predicted interactions are indicated by lines with different colors.

**Table 1 ijms-24-15005-t001:** Characterization of *SlMLP* genes in tomato.

Gene Name	Gene ID	CDS Size	Protein			GRAVY	Instability Index	Subcellular Localization Prediction
		(bp)	Length (aa)	MW (kDa)	pI			
SlMLP1	Solyc03g117450.1	480	159	17.61	5.02	−0.350	32.81	Cytoplasm
SlMLP2	Solyc03g117460.1	480	159	17.64	4.90	−0.401	34.90	Cytoplasm
SlMLP3	Solyc04g007010.3	441	146	16.61	5.96	−0.168	33.78	Cytoplasm
SlMLP4	Solyc04g007750.4	498	165	18.93	5.85	−0.256	34.38	Cytoplasm
SlMLP5	Solyc04g007760.3	441	146	16.58	6.03	−0.200	25.97	Cytoplasm
SlMLP6	Solyc04g150102.1	441	146	16.57	5.07	−0.147	41.68	Cytoplasm
SlMLP7	Solyc04g007770.3	444	147	16.60	5.96	−0.162	42.21	Cytoplasm
SlMLP8	Solyc04g007780.3	447	148	17.02	5.62	−0.403	19.96	Cytoplasm
SlMLP9	Solyc04g007790.3	444	147	16.82	5.63	−0.407	17.31	Cytoplasm
SlMLP10	Solyc04g150104.1	891	296	34.40	5.10	−0.429	39.14	Cytoplasm, Cytoskeleton
SlMLP11	Solyc04g007820.3	444	147	16.77	5.72	−0.365	35.06	Nucleus
SlMLP12	Solyc04g007825.2	555	179	20.96	5.79	−0.411	35.98	Cytoplasm
SlMLP13	Solyc04g050950.3	444	147	17.15	5.90	−0.351	36.51	Cytoplasm
SlMLP14	Solyc05g005865.1	462	153	17.09	5.17	−0.231	24.73	Extracellular
SlMLP15	Solyc05g046140.3	453	150	17.60	5.22	−0.467	50.34	Cytoplasm
SlMLP16	Solyc05g046150.3	453	150	17.70	5.47	−0.456	51.29	Cytoplasm
SlMLP17	Solyc05g054380.2	489	162	18.18	4.78	−0.188	37.17	Cytoplasm
SlMLP18	Solyc07g005370.4	501	166	18.82	5.48	−0.150	25.62	Cytoplasm
SlMLP19	Solyc07g008710.3	480	159	18.42	5.31	−0.613	32.72	Cytoplasm
SlMLP20	Solyc08g023660.3	447	148	17.07	6.50	−0.420	16.91	Cytoplasm
SlMLP21	Solyc09g005400.3	453	150	17.07	5.17	−0.187	33.43	Cytoplasm
SlMLP22	Solyc09g005420.4	510	169	19.39	5.29	−0.166	28.23	Extracellular
SlMLP23	Solyc09g005425.1	456	151	17.43	5.10	−0.438	40.68	Extracellular
SlMLP24	Solyc09g005500.3	456	151	17.43	5.10	−0.438	40.68	Extracellular
SlMLP25	Solyc09g014525.1	453	150	17.10	5.57	−0.223	25.66	Cytoplasm
SlMLP26	Solyc09g014550.3	459	152	17.41	5.47	−0.266	24.29	Cytoplasm
SlMLP27	Solyc09g014580.3	459	152	17.42	5.62	−0.248	31.78	Cytoplasm
SlMLP28	Solyc09g090970.4	552	183	20.55	6.41	−0.322	25.35	Cytoplasm
SlMLP29	Solyc09g090980.3	483	160	17.37	5.44	−0.128	30.62	Cytoplasm
SlMLP30	Solyc09g090990.2	483	160	17.91	5.34	−0.426	32.81	Cytoplasm
SlMLP31	Solyc09g091000.4	552	183	20.76	6.38	−0.383	35.34	Cytoplasm
SlMLP32	Solyc10g008330.4	474	157	18.29	8.73	−0.718	25.59	Cytoplasm
SlMLP33	Solyc10g048030.2	441	146	16.69	6.37	−0.305	13.79	Cytoplasm
SlMLP34	Solyc12g096960.2	477	158	18.20	5.03	−0.218	38.70	Cytoplasm

## Data Availability

All data are provided within the main text and Appendix A.

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
