# Peer review of "Genome-Wide Evolutionary Characterization and Expression Analysis of Major Latex Protein (MLP) Family Genes in Tomato"

_ijms, 2023, doi:10.3390/ijms241915005_

Round 1
Reviewer 1 Report
The Manuscript entitled "Genome-wide evolutionary characterization and expression analysis of MLP family genes in tomato" is written well and provides good starting point to explore the roles of MLPs in plants. I have few comments about the MS.
1) The introduction part is good. It clearly tells about the background and aim of the study.
2) Results are fine and correlate with the discussion section.
3) There are minor typo errors. For eg. lines 352, 495 & 496. Go through the MS again and correct them.
4) In line 406, there should be Fig. S4 for venn diagram instead of Fig. S3. As supplementary file shows Fig. S4 is for venn diagram.
5) It looks more presentable if you can provide one more heat map/venn diagram which can summarize the qRTPCR results of SlMLP genes in abiotic stresses.
Reviewer 2 Report
Dear Authors,
Authors presented complex research concentrated on genome-wide search and identification of tomato MLP gene family members. Moreover, analyses were performed by combining bioinformatics tools and molecular assays to analyze their phylogenetic relationships, gene structure, chromosome distribution and also localization, conserved structural domains and patterns, and expression of MLP family members. Unfortunately, Authors have an real problem with obtained data interpretation;
There are some the most important aspects that need to be clarified or improved:
-It can be find factual error- Authors stated: „expression of MLP family 73 members in different tissues of tomato, including roots, stems, leaves, flowers, fruits, and 74 seeds” – Roots, leaves or seeds are not plant’s tissues !
-figure 1 should be improved to the more readable;
-similar situation with figure 2 – it is almost unreadable, maybe it should be separate (?)
-figure 5 should be improved definitely;
- as I state before, Authors did not presented tissue gene expression; And it is a very good question in what kind of tissues and compartments presented genes can be expressed ?
- figure 7 should be separated to the panel because all chart are unreadable;
-Authors did not recognize the cellular compartments, therefore they did not presented GFP signal in cytoplasm; Moreover, information, that the analyses was performed only in leaf epidermis should be added definitely;
- materials and methods are quite good described, but information about two used reference genes are needed (without searching in supplementary tables with primers);
Sincerely
Some moderate English correction can be a good idea to improved the manuscript;
Round 2
Reviewer 2 Report
Authors significantly improved the manuscript, especially data concernings localisations apects; Unfortunately Figure 2 is still very difficult to read- I suggest to improve it more extensively;
Most of Author's explanations are clear;